# Increased Thyroidal Activity on Routine FDG-PET/CT after Combination Immune Checkpoint Inhibition: Temporal Associations with Clinical and Biochemical Thyroiditis

**DOI:** 10.3390/cancers15245803

**Published:** 2023-12-11

**Authors:** Anna Galligan, Roslyn Wallace, Balasubramanian Krishnamurthy, Thomas W. H. Kay, Nirupa Sachithanandan, Cherie Chiang, Shahneen Sandhu, Rodney J. Hicks, Amir Iravani

**Affiliations:** 1Department of Endocrinology and Diabetes, St Vincent’s Hospital Melbourne, Melbourne, VIC 3065, Australia; bmurthy@svi.edu.au (B.K.); tkay@svi.edu.au (T.W.H.K.); nirupa.sachithanandan@svha.org.au (N.S.); 2Department of Medicine, St Vincent’s Hospital Medical School, University of Melbourne, Melbourne, VIC 3010, Australia; rod.hicks@premit.net.au; 3Immunology and Diabetes Unit, St. Vincent’s Institute of Medical Research, Fitzroy, VIC 3065, Australia; 4Department of Medical Oncology, Peter MacCallum Cancer Centre, Melbourne, VIC 3052, Australia; roslyn.wallace@petermac.org (R.W.); shahneen.sandhu@petermac.org (S.S.); 5Department of Internal Medicine, Peter MacCallum Cancer Centre, Melbourne, VIC 3052, Australia; cherie.chiang@mh.org.au; 6Department of Medicine, Royal Melbourne Hospital Medical School, University of Melbourne, Melbourne, VIC 3010, Australia; 7The Sir Peter MacCallum Department of Oncology, University of Melbourne, Melbourne, VIC 3010, Australia; 8Department of Cancer Imaging, Peter MacCallum Cancer Centre, Melbourne, VIC 3000, Australia; airavani@uw.edu; 9Department of Radiology, University of Washington, Seattle, WA 98195, USA

**Keywords:** combination immunotherapy, ipilimumab and nivolumab, immune-related adverse event, endocrine toxicity, thyroiditis, FDG-PET/CT, interobserver agreement, melanoma

## Abstract

**Simple Summary:**

Thyroiditis is one of the most common immune toxicities in patients treated with combination immune checkpoint inhibition. Increased metabolic activity in the thyroid can be observed on FDG-PET/CT performed for the purposes of response assessment. The aim of our retrospective study was to assess the association and temporal profile of FDG-PET/CT findings suggesting thyroiditis with the natural history of clinical and biochemical thyroid dysfunction. Our findings demonstrated that an increased metabolic activity of the thyroid seen on routine FDG-PET/CT correlates with a biochemical diagnosis of thyroiditis and persists well after the initial biochemical diagnosis. We demonstrated that patients with FDG-PET/CT changes were more likely to develop overt clinical hypothyroidism.

**Abstract:**

Background: FDG-PET/CT used for immune checkpoint inhibitor (ICI) response assessment can incidentally identify immune-related adverse events (irAEs), including thyroiditis. This study aimed to correlate the time course of FDG-PET/CT evidence of thyroiditis with clinical and biochemical evolution of thyroid dysfunction. Methods: A retrospective review was performed by two independent blinded nuclear medicine physicians (NMPs) of thyroidal FDG uptake in 127 patients who underwent PET/CT between January 2016 and January 2019 at baseline and during treatment monitoring of combination ICI therapy for advanced melanoma. Interobserver agreement was assessed and FDG-PET/CT performance defined by a receiver-operating characteristic (ROC) curve using thyroid function tests (TFTs) as the standard of truth. Thyroid maximum standardized uptake value (SUVmax) and its temporal changes with respect to the longitudinal biochemistry were serially recorded. Results: At a median of 3 weeks after commencing ICI, 43/127 (34%) had a diagnosis of thyroiditis established by abnormal TFTs. FDG-PET/CT was performed at baseline and at a median of 11 weeks (range 3–32) following the start of therapy. ROC analysis showed an area under the curve of 0.87 (95% CI 0.80, 0.94) for FDG-PET/CT for detection of thyroiditis with a positive predictive value of 93%. Among patients with biochemical evidence of thyroiditis, those with a positive FDG-PET/CT were more likely to develop overt hypothyroidism (77% versus 35%, *p* < 0.01). In the evaluation of the index test, there was an almost perfect interobserver agreement between NMPs of 93.7% (95% CI 89.4–98.0), kappa 0.83. Conclusion: Increased metabolic activity of the thyroid on routine FDG-PET/CT performed for tumoral response of patients undergoing ICI therapy is generally detected well after routine biochemical diagnosis. Elevation of FDG uptake in the thyroid is predictive of overt clinical hypothyroidism and suggests that an ongoing robust inflammatory response beyond the initial thyrotoxic phase may be indicative of thyroid destruction.

## 1. Introduction

Combination immune checkpoint inhibition (cICI) with cytotoxic T-lymphocyte associated protein 4 (CTLA-4) and programmed cell death protein 1 (PD-1) or its ligand (PD-L1) inhibition is approved for use in patients with advanced melanoma and other malignancies including renal cell carcinoma and non-small cell lung cancer [1,2,3,4]. The improved survival benefit in these previously poor prognosis malignancies with combination therapy is offset by an increase in incidence and severity of collateral immune toxicity to healthy organs, collectively termed immune-related adverse events (irAEs) [2,5]. One of the most common irAEs after any ICI is thyroiditis [6,7]. In monotherapy clinical trials, thyroiditis was more common after blockade of the PD-1/PD-L1 pathway with an incidence of 8–38% compared to CTLA-4 inhibition with an incidence of 6–25% [7,8,9]. With the combination of ipilimumab and nivolumab in the landmark Checkmate 067 trial, hyper and/or hypothyroidism occurred in 28% [4]. Since then, observational studies have reported an incidence of thyroiditis after cICI of up to 56% [7,10].

In contrast to other forms of spontaneous thyroid autoimmunity, ICI therapy appears to induce a rapid onset, inflammatory thyroiditis resulting in a leak of pre-formed thyroid hormone and varying degrees of thyroid destruction [11]. Subclinical thyrotoxicosis, overt thyrotoxicosis and hypothyroidism are independently reported in clinical trial safety outcomes, but most probably reflect different time points of a common disease trajectory [12]. Hypothyroidism without a preceding thyrotoxic phase is less common [7].

At our center, ^18^F-fluorodeoxyglucose positron emission tomography/computed tomography (FDG-PET/CT) is routinely utilized for tumor response assessment in patients receiving cICI for advanced melanoma [13,14]. A limited number of studies indicate that new increased metabolic activity in specific organs, including the thyroid gland, on routine FDG-PET/CT imaging performed in the first 3–6 months of treatment may be suggestive of irAEs [15,16,17,18]. While the finding of increased uptake of FDG in the thyroid after ICI is common [19], the temporal relationship between these findings and clinical and biochemical evidence of thyroiditis has not been systematically evaluated. We aimed to determine the association of FDG-PET/CT findings suggestive of thyroiditis developing in response to cICI as compared to the reference standard of biochemical assessment of thyroid function tests (TFTs). We further sought to evaluate interobserver agreement for FDG-PET/CT as a novel modality for the detection of ongoing thyroiditis and derive semiquantitative cut-offs values for this diagnosis.

## 2. Methods and Analysis

A pharmacy dispensing report was used to identify all patients who received at least one dose of cICI for metastatic melanoma between January 2016 and January 2019 at The Peter MacCallum Cancer Centre; a quaternary melanoma center in Melbourne, Victoria, Australia. Patients were eligible for inclusion if they had available FDG-PET/CT imaging at baseline and within six months of commencing cICI. Patients who had received prior first-line therapy with single-agent anti-PD-1, anti-CTLA-1 or BRAF/MEK inhibition were included. However, patients with a documented history of thyroiditis related to a prior line of immunotherapy were excluded. A chart review was undertaken to identify patients diagnosed with thyroiditis and the corresponding clinical presentation, TFT results and thyroid autoantibodies were recorded. Clinical events were followed for a minimum of 12 months.

### 2.1. Test Methods

#### 2.1.1. Reference Standard

In keeping with the American Society of Clinical Oncology (ASCO) guidelines, patients receiving cICI had TFTs at baseline, at every cycle for the first four cycles, every 4–6 weeks thereafter, or earlier in the case of clinical suspicion. TFTs included thyroid stimulating hormone (TSH), free thyroxine (FT4) and free triiodothyronine (FT3) and were performed on the Abbot Architect chemiluminescent immunoassay (Abbott Diagnostics, Abbot Park, IL). In-house laboratory pre-specified population reference ranges were utilized for TSH (0.35–4.94 mU/L), FT4 (9–19 pmol/L) and FT3 (2.6–5.7 pmol/L). Biochemical thyroiditis was defined as (1) subclinical thyrotoxicosis (low TSH, normal FT3 and FT4 in the absence of suspected hypophysitis), (2) overt thyrotoxicosis (low TSH, elevated FT3 and/or FT4), (3) subclinical hypothyroidism (high TSH, normal FT3 and FT4) or (4) overt hypothyroidism (high TSH, low FT3 or FT4). Time to diagnosis of thyroiditis, duration of the thyrotoxic phase and percentage developing permanent hypothyroidism were retrospectively assessed. When measured, levels of anti-thyroperoxidase antibodies (TPOAb) and anti-thyroglobulin antibodies (TgAbs) were performed on an electrochemiluminescent immunoassay (Elecsys Anti-TPO kit and Elecsys Anti-Tg kit, respectively; Roche Diagnostics, Mannheim Germany). All clinical and biochemical information was extracted by an endocrinologist (AG), blinded to FDG PET/CT analysis.

#### 2.1.2. Index Test

All studies were performed on an integrated PET/CT scanner, including Biograph 16 (Siemens Medical Solutions, Erlangen, Germany), GE Discovery 690 or GE Discovery 710 (GE Healthcare, Milwaukee, WI, USA), with routine cross-calibration three-monthly. FDG-PET/CT was performed as per European Association of Nuclear Medicine Research Ltd. (EARL) initiative [20]. Contrast enhancement was not used for the CT component. Although there are no established criteria for the diagnosis of thyroiditis based on FDG-PET/CT, a normal thyroid demonstrates no increased metabolic activity above the background and should not be seen as a discrete organ on PET images. Therefore, a visible thyroid on FDG-PET/CT images, which was new compared to the pre-treatment study, was considered suggestive of thyroiditis. Two experienced nuclear medicine physicians (NMPs, AI and RJH) blinded to clinical data and the reference standard results assessed all available FDG-PET/CT scans independently. Majority agreement was used to define findings suggestive of thyroiditis. If only one NMP noted an increase in FDG uptake, the assessment was considered negative for thyroiditis. The interobserver agreement for FDG-PET/CT interpretation of thyroiditis was evaluated. In addition, a semiquantitative analysis on PET images was performed by a 1 cm^3^ spherical volume of interest over the thyroid gland, as identified on the CT portion of PET/CT, using MIM software (MIM 6.7.11; MIM Software, Cleveland, OH, USA). The maximum standardized uptake value (SUVmax) in the thyroid gland was measured at baseline, after initiation of cICI and until resolution and percentage increase in SUVmax from baseline was compared.

### 2.2. Analysis

The reference standard for a diagnosis of thyroiditis was determined by TFTs. Clinical variables in patients with and without thyroiditis were compared using Fisher’s exact, χ^2^ and Wilcoxon rank sum tests. Time to diagnosis of thyroiditis and duration of the thyrotoxic phase were described as continuous variables with median and interquartile ranges (IQRs). The duration of the thyrotoxic phase was defined as the time in weeks from first TFT evidence of subclinical or overt thyrotoxicosis to normalization of TFTs or a TSH above the upper limit of normal (whichever occurred first). Patients diagnosed with thyroiditis who had indeterminate follow-up results were excluded from the analysis of the duration of the thyrotoxic phase.

The result of the index test was listed as a binary positive or negative. For the purpose of the analysis, equivocal results (one user only) were classified as negative. The diagnostic accuracy of FDG-PET/CT with respect to the reference standard was represented as sensitivity, specificity, positive predictive value (PPV) and negative predictive value (NPV) with 95% confidence intervals. The AUC for the overall performance of FDG-PET/CT was calculated as the mean of the estimated sensitivity and specificity by reported operating characteristics (ROC). Interobserver agreement for the diagnosis of thyroiditis by FDG-PET was assessed by the Cohen’s kappa statistics. The optimal cut-offs for thyroid SUVmax and its temporal changes suggestive of thyroiditis were selected based on a predictive model for optimal sensitivity and specificity. All statistical analyses were performed using STATA version 16.1 (STATA LP, College Station, TX, USA).

## 3. Results

### 3.1. Participants

Of 162 patients treated with combination ICIs from January 2016 to January 2019, 127 patients met the inclusion criteria for the study. Of the 35 patients excluded, 28 lacked post-treatment FDG-PET imaging and seven patients had a prior history of thyroiditis. The median age was 59 years (range 18–79) and 29% were female. Overall, 43/127 (34%) had a diagnosis of thyroiditis established with TFTs (Figure 1). There was no difference in patient demographics between those with and without thyroiditis (Table 1). Prior exposure to anti-CTLA4 or anti-PD1 was not associated with new-onset thyroiditis after cICI. A higher proportion of patients with thyroiditis had previously received BRAF/MEK inhibition (58% versus 31%, *p* = 0.003).

### 3.2. Established Thyroiditis

Thyroiditis was diagnosed biochemically at a median of 3 weeks from commencement of cICI (range 1–15). 37/43 (86%) patients had a documented thyrotoxic phase, which was overt in 34 and subclinical in three, and lasted a median duration of 4.5 weeks. The thyrotoxic phase was transient in all cases. Six patients first presented with hypothyroidism, and 19/37 (51%) patients with a transient thyrotoxic phase progressed to a hypothyroid phase. Overall, thyroxine replacement was commenced in 23/43 patients. Once commenced, thyroxine withdrawal was not attempted by the treating clinicians. Most patients (38/43) were managed in the outpatient setting with mild symptoms. Five patients required hospital admission during the thyrotoxic phase, although four were primarily receiving treatment for another contemporaneous irAE. Two patients complained of mild transient neck pain which did not require treatment with glucocorticoids. TPO and TG ab were measured in 17/43 patients and were elevated in six (35%) and 11 (65%) respectively. TSHrAb was measured in 13 people. Only one individual had a high titre of TSHrAb (>40 IU/L, reference range < 1.8 IU/L). Thyroid hormone levels in this individual followed a pattern of minimally symptomatic transient thyrotoxicosis followed by permanent hypothyroidism. In this case, the presence of TSHrAb did not herald prolonged thyroid hormone overproduction.

### 3.3. Index Test Results

FDG/PET-CT was performed in all patients at a median of 11 weeks from commencement of cICI (range 3–32). FDG/PET-CT was declared positive by both NMPs in 26/43 (60%) patients with thyroiditis but only 2/84 (2%) patients not diagnosed with thyroiditis (Table 2), resulting in an AUC of 0.87 [95% CI 0.8, 0.94]. Recognizing the temporal discordance in biochemical testing and scanning in many patients, the sensitivity, specificity, positive predictive value (PPV) and negative predictive value (NPV) were 61% [95% confidence interval (CI) 44–75], 98% [95% CI 92–100%], 93% [95% CI 77–91] and 83% [95% CI 74–90], respectively.

Temporal evolution of the thyroid FDG uptake in a patient with thyroiditis is demonstrated in Figure 2. In 22/28 patients with PET-thyroiditis and available further follow-up FDG PET/CT, resolution of FDG uptake to baseline was noted in all cases by a median of 28.6 weeks (range 17.9–158) from the start of cICIs.

#### 3.3.1. Interobserver Agreement

Both users agreed on 93.7% of index test results [95% CI 89.4–98.0] (Table 3). The Cohen’s kappa score was 0.83, indicating almost perfect interobserver agreement between both users according to the scale provided by Landis and colleagues [21].

#### 3.3.2. Semiquantitative Analysis

The median baseline thyroid SUVmax in the cohort was 1.8 (IQR 1.5–2.1). When both users agreed on the qualitative diagnosis of thyroiditis, the median SUVmax at the first follow-up scan was 5.15 (IQR 3.95–6) with a corresponding absolute and % increase of 2.7 (IQR 1.9–3.8) and 127% (IQR 89–190%) respectively. In patients with equivocal results (one user only), the median SUVmax at first follow-up was 2.8 (IQR 2.45–2.95), with a corresponding absolute and % increase of 1.1 (IQR 0.35–1.35) and 61% (IQR 18–88%) respectively. When both users agreed on a negative result, the median SUVmax at first follow-up was 1.6 (IQR 1.4–1.9) with a median change of 0% (IQR -16–10%) (Figure 3).

Using a predictive model, the SUVmax cut-off of 2.8 infers a sensitivity of 71%, with a specificity of 95% and correctly classifies thyroiditis in 87% of patients. The SUVmax (% change) cut-off of 52% infers a sensitivity of 71%, specificity of 98% and correctly classifies thyroiditis in 89% of patients. These cut-offs increase the overall test sensitivity from 61% with a small decrease in specificity for SUVmax from 98% but no change to specificity using % change. The ROCs for SUVmax and % change are presented in Figure 4.

### 3.4. Clinical Correlation

The timing of FDG-PET/CT with respect to the biochemical diagnosis is illustrated in Table 4 and Figure 5. In almost all cases, the biochemical diagnosis preceded the first on treatment FDG-PET/CT. There was no difference in the time interval between biochemical thyroiditis and the first FDG-PET/CT in patients with and without FDG-PET-detected thyroiditis.

Overall, patients with FDG-PET-detected thyroiditis were more likely to develop overt hypothyroidism (71% versus 6%, *p* < 0.001). One patient had an abnormal diffuse thyroid uptake on FDG-PET at baseline (2 months prior to cICI), which resolved on the first follow up study (Figure 6). This patient had normal thyroid function at the time of the baseline FDG-PET but developed severe thyrotoxicosis (FT4 138 pmol/L, TSH < 0.01 mU/L) 10 days after commencing cICI, 2 months after the abnormal baseline FDG-PET/CT and 1.5 months before a normal follow up FDG-PET/CT. Autoantibodies were not measured. No prior anti-PD-1 or anti-CTLA-4 therapy had been administered in this case, indicating a probable pre-existing thyroid autoimmunity.

There were two patients without clinical thyroiditis who had an abnormal FDG-PET/CT result. One patient was diagnosed with hypophysitis and secondary hypothyroidism. While there was no thyrotoxic phase, subsequent primary hypothyroidism could not be excluded. The second patient had normal TFTs two weeks before and one month after the abnormal FDG-PET and remained euthyroid.

### 3.5. Concomitant Muscle Uptake

In the 28 patients with FDG-PET/CT-detected thyroiditis, there were two patients with overt evidence of FDG-PET manifestations of myopathy. In both cases, transient diffuse skeletal muscle FDG uptake occurred concurrently with increased thyroid uptake and biochemical thyroiditis, suggesting the differential diagnoses of thyroid myopathy or concomitant inflammatory myositis (Figure 7) One patient remained asymptomatic while the other experienced significant proximal weakness for the duration of the thyrotoxic phase. This patient developed biochemical thyrotoxicosis 9 days after commencement of cICI, while receiving high-dose glucocorticoid treatment for cerebral metastases. Over the following 3 weeks, progressive generalized muscle weakness was reported despite a decreasing dose schedule of glucocorticoids. Random cortisol and serum creatine kinase levels were normal. FDG-PET/CT performed two months after treatment commencement demonstrated interval development of thyroid and diffuse skeletal muscle uptake. The weakness resolved after transition to the hypothyroid phase.

## 4. Discussion

Our data show that cICI-related thyroiditis was common, with a prevalence similar to the combination arm of the CheckMate 067 study (34% compared to 28%) [2]. FDG-PET/CT had a high specificity and positive predictive value for thyroiditis with respect to the reference standard biochemical test, although the temporal profiles of biochemical and scan findings were out of sync, primarily related to the discordant timing of routine biochemical testing and FDG PET/CT response assessment. Biochemical diagnosis of thyroiditis occurred early, at a median of 3 weeks from commencement of cICI. The lower sensitivity of FDG-PET/CT may relate to the timing of the imaging with respect to onset of the thyroiditis. With the median time of the first FDG-PET/CT scan 11 weeks, most patients had already developed biochemical abnormality and, in a subgroup of patients, the acute inflammatory phase (as represented by increased FDG uptake in the gland) may have resolved by the time of the first scan.

As changes in thyroidal uptake occur after the diagnosis is made by biochemical thyroid function tests, the incidental finding of diffusely increased FDG uptake on routine on-treatment FDG PET/CT is highly diagnostic of thyroiditis rather than thyroidal tumor metastases.

The initial thyrotoxic phase was generally detected prior to FDG-PET/CT evaluation and was largely resolved by the time FDG-PET/CT demonstrated evidence of ongoing increasing FDG uptake. This finding suggests that thyroid inflammation continues beyond the phase of initial thyroiditis with hormone release and may indicate an ongoing process of thyroid destruction.

Our finding that 53.5% of patients developed permanent hypothyroidism is similar to the published literature. In one study of 1246 patients treated with checkpoint inhibitors, permanent hypothyroidism requiring levothyroxine replacement occurred in 43% of patients who initially presented with overt thyrotoxicosis and only 3% of those whose initial presentation was subclinical thyrotoxicosis [22].

Indeed, patients who had FDG-PET/CT evidence of thyroiditis had a higher likelihood of developing overt hypothyroidism requiring ongoing thyroid hormone replacement. Hence, we postulate that the presence of increased FDG activity may indicate a more persistent and robust autoimmune response with destruction of thyrocytes by the adoptive immune system. Whether thyroid hormone replacement could be withdrawn when FDG uptake in the gland had resolved was not tested in this population, with clinicians continuing treatment once initiated. This practice is in line with the literature, which suggests most cases of overt hypothyroidism after ICI are permanent [7].

### 4.1. Pathophysiology of Thyroiditis

Currently, there are very little data as to the pathophysiology of ICI-related thyroiditis. As a glucose analogue radiotracer, FDG uptake in thyroiditis may reflect the increase glycolytic activity of activated CD8+ T cells infiltrating the gland [23]. Cytological and molecular analyses of thyroid tissue would be of benefit in understanding the underlying mechanisms. However, thyroid biopsy in otherwise recovering patients is difficult to justify [6]. In our cohort, TPOAb and TgAb were elevated in 35% and 65%, respectively, compared to the expected 95% and 80% reported in spontaneous autoimmune thyroid disease [24]. One observational study has shown that TgAb but not TPOAb positivity is associated with an increased likelihood of progression to permanent hypothyroidism following the hyperthyroid phase [25]. In our study, baseline thyroid antibody levels were not measured. Muir et al. have demonstrated across several studies that baseline TPOAb and TGAb positivity were more common in patients who developed thyroid irAEs than in those who did not, and patients where there was a greater than 50% increase in baseline antibody titre or new seroconversion were similarly more likely to experience thyroiditis than those who did not [22,25]. In thyroid irAE, antibody seroconversion may be a late or bystander response to a rapid pathogenic T cell autoimmunity, or antibody specificity in thyroid irAE may be different to that of spontaneous thyroiditis. The heterogeneity of the antibody response after treatment likely negates them as a clinically useful biomarker, however baseline antibody levels may be useful for risk prediction.

New onset Graves’ disease after ICI is rare but has been described in case reports, and guidelines do recommend measuring TSHrAb in any case of severe or symptomatic thyrotoxicosis to guide initiation of antithyroid drugs [26,27,28]. The one patient with a high titre of TSHrAb in our cohort experienced a brief thyrotoxic phase followed by permanent hypothyroidism. The absence of evidence of thyroid hormone overproduction in ICI-related thyroiditis prompts the recommendation that anti-thyroid drugs are unlikely to be useful in this setting. If commenced due to clinical concern or an elevated TSHrAb, TFTs should be monitored within 2-weekly intervals with a low threshold to cease treatment once the spontaneous thyrotoxic phase resolves.

### 4.2. FDG-PET/CT in ICI Related Thyroiditis

Incidental diffuse thyroid uptake on FDG-PET/CT is known to be associated with a higher incidence of spontaneous thyroid dysfunction in adults free of malignancy [29]. A limited number of studies have reviewed the role of FDG-PET/CT in ICI-related thyroiditis. At our center, we have previously demonstrated the accuracy of FDG-PET/CT in detecting both tumoral and systemic immune response. In our prior study, increased uptake in the thyroid following treatment was found in six patients, all of whom had biochemical correlation [15]. Yamauchi and colleagues reviewed results from baseline FDG-PET/CT imaging performed prior to ICI administration in 111 patients. A significant correlation between increased thyroid uptake before treatment and development of thyroiditis after treatment was observed (46.7% versus 4.2% *p* < 0.001) [16]. In our current study, all patients had TFTs prior to cICI and we excluded patients with a prior history of thyroid dysfunction. However, we did find evidence of pre-treatment FDG-PET/CT-detected thyroiditis in one patient who went on to develop new thyrotoxicosis soon after the first dose of cICI. In a small study of 18 lung cancer patients treated with nivolumab, baseline imaging was not associated with thyroiditis, though increased uptake in the thyroid 10–16 weeks after treatment commencement was predictive of hypothyroidism before the TSH became elevated [19]. In that study, only six patients had thyroiditis. The finding that the presence of thyroid metabolic activity on FDG-PET/CT is predictive of overt hypothyroidism was replicated in our study with a larger number of patients.

Because FDG-PET/CT imaging is commonly performed at an interval of 12 weeks from the onset of cICI largely to assess tumoral response and biochemical thyroiditis occurs within three weeks of commencing cICI, FDG-PET/CT findings suggestive of thyroiditis have limited diagnostic clinical utility. Nonetheless, an incidental observation of increased thyroid uptake should prompt careful monitoring of thyroid function if not already known to be abnormal. Whether thyroid hormone replacement can be safely withdrawn after resolution of active thyroid uptake of FDG on PET remains unknown.

### 4.3. Increased Incidence of Thyroiditis in Patients Previously Exposed to BRAF/MEK Inhibition

Our finding of a higher proportion of thyroiditis in patients who previously received BRAF/MEK inhibition (58% versus 31%, *p* = 0.003) has to our knowledge not been previously reported. Mitogen-activated protein kinase (MAPK) pathway signaling is important to thyroid biology as demonstrated in thyroid cancer. In papillary thyroid carcinoma driven by BRAF pathway mutations, there is decreased expression of sodium iodide symporter (NIS), TSH receptors and tumor cell specific MHC II. MHC class II expression in tumor cells increases antigen presentation to prime infiltrating CD^4+^ T cells, and decreased tumor cell MHC class II expression may result in immune escape. These proteins are restored by BRAF pathway inhibitors and thus enhancing immune response to tumor cells. It is not known if thyroid cell MHC class II expression is upregulated in normal thyroid cells to explain the increased immune response against thyroid cells following treatment with BRAF inhibitors seen in this cohort [30,31]. Perhaps inhibition of this pathway may pre-sensitize the thyroid to expressing alterative cell membrane autoantigens. Of note, prior treatment with CTLA-4 inhibitors or PD-1 inhibitors did not appear to prime the thyroid for thyroiditis events after cICI.

## 5. Conclusions

The primary role of FDG-PET/CT is monitoring of tumoral response and not for the evaluation of thyroiditis. While we admit the limitations in the utility of FDG PET/CT in the initial diagnosis of thyroiditis, the study has followed the recommendations for novel test performance characteristics in the analysis of the sensitivity/specificity/NPV/PPV of FDG PET/CT with respect to the reference standard test. With an increasing incidence of thyroid abnormalities detected on routine FDG-PET/CT imaging it is of great importance to understand the implications of these findings when interpreting the FDG PET/CT. To our knowledge, this is the first study to systematically evaluate the temporal profile of FDG-PET/CT findings of thyroiditis in relation to the biochemical and clinical course of thyroid dysfunction after cICI.

We demonstrate that biochemical thyrotoxicosis and hypothyroidism usually represent different time-points along the same disease trajectory, but a minority present with hypothyroidism not preceded by a thyrotoxic phase. Although routine FDG-PET/CT is commonly performed after the biochemical thyroid abnormality was detected, our study demonstrates that FDG-PET/CT has a high specificity and positive predictive value for ICI-related thyroiditis and correlates with progression to permanent hypothyroidism. We demonstrate high interobserver agreement for the detection of thyroiditis with FDG-PET and describe semiquantitative cut-off values using SUVmax and % change with respect to baseline.

## Figures and Tables

**Figure 1 cancers-15-05803-f001:**
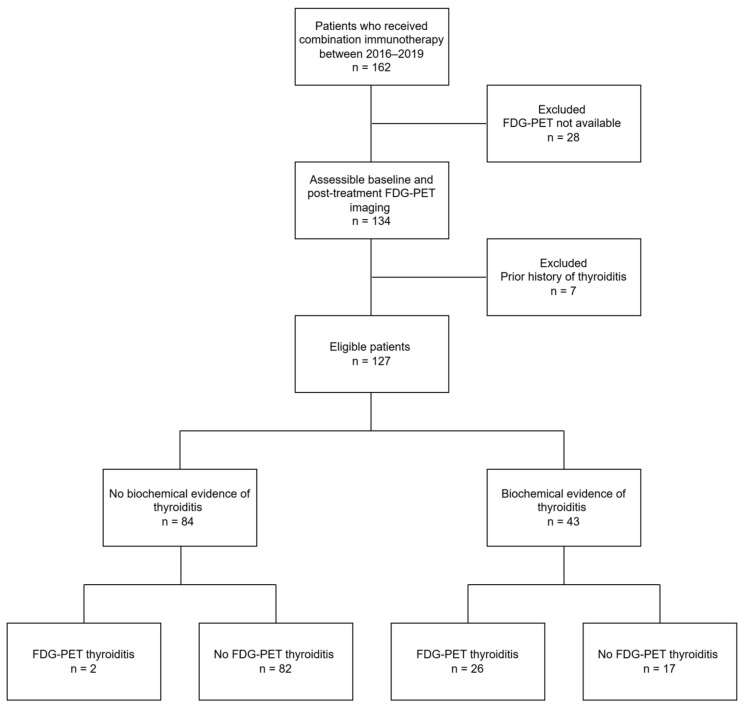
Consort diagram outlining patient eligibility and summary of FDG-PET/CT and clinical findings.

**Figure 2 cancers-15-05803-f002:**
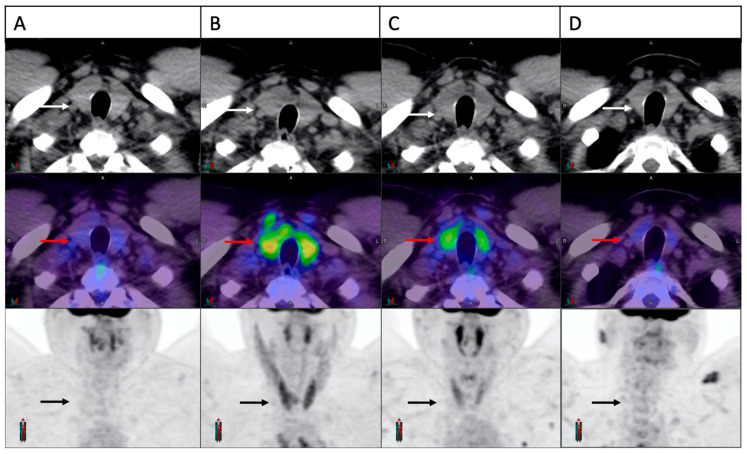
Temporal evolution of thyroid FDG uptake in a patient with cICI related thyroiditis. Trans-axial and maximum intensity projection images of the thyroid gland demonstrated in sequence on CT scan (white arrows), fusion image (red arrows) and PET scan (black arrows). Normal thyroid appearance is demonstrated at baseline (**A**). A significant increase in FDG uptake is noted on the first on-treatment scan at three weeks (**B**). Panel (**C**,**D**) demonstrate partial and complete resolution at 6 weeks and 24 weeks, respectively.

**Figure 3 cancers-15-05803-f003:**
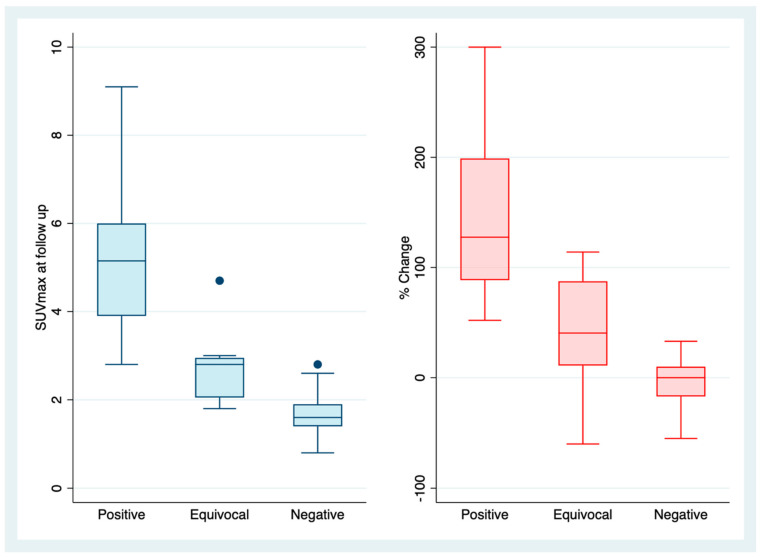
Box plot SUVmax (blue) and % change of SUV max (red). Absolute SUVmax at the first on-treatment FDG-PET/CT, and % change in SUVmax with respect to baseline in patients where FDG-PET/CT-detected thyroiditis was documented by both NMPs (positive), by one NMP (equivocal) and by neither NMP (negative). Equivocal results were deemed to reflect nuanced reporting in the lower range of abnormality and were classified as negative in the analysis.

**Figure 4 cancers-15-05803-f004:**
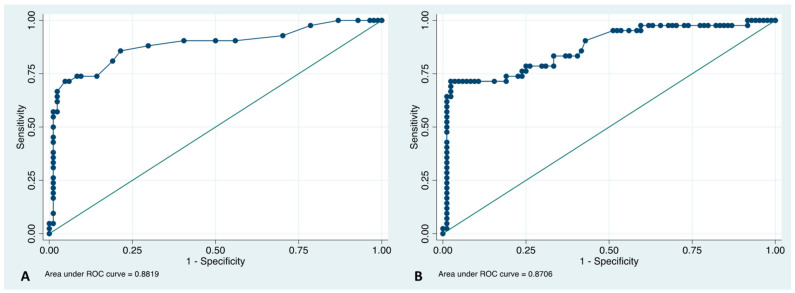
Receiver Operator Curves (ROC). (**A**) SUVmax at first follow up scan and (**B**) SUVmax% change with corresponding area under the ROC of 0.88 and 0.87, respectively.

**Figure 5 cancers-15-05803-f005:**
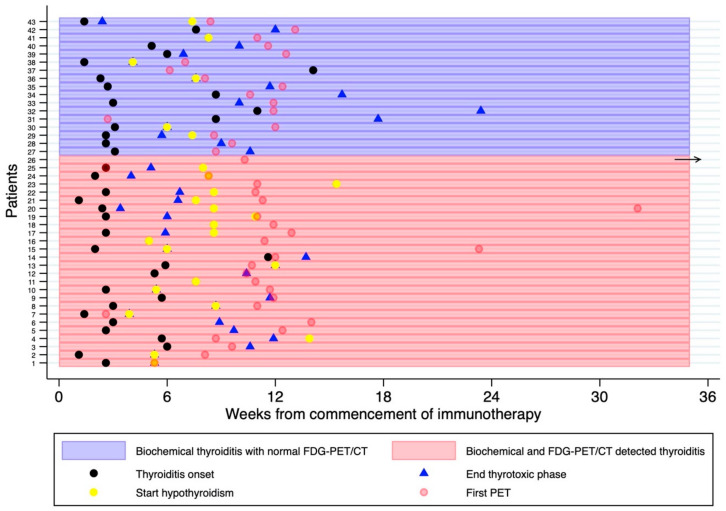
Swimmer Plot demonstrating the timing of the first on-treatment FDG-PET/CT with respect to the thyroiditis diagnosis, typically represented by an early thyrotoxic phase followed by euthyroid and/or hypothyroid phase. Thyroid function tests were performed every 4–6 weeks or more frequently if clinically indicated. The horizontal black arrow represents an outlier, whereby the thyrotoxicosis was diagnosed 97 months after the first FDG-PET/CT.

**Figure 6 cancers-15-05803-f006:**
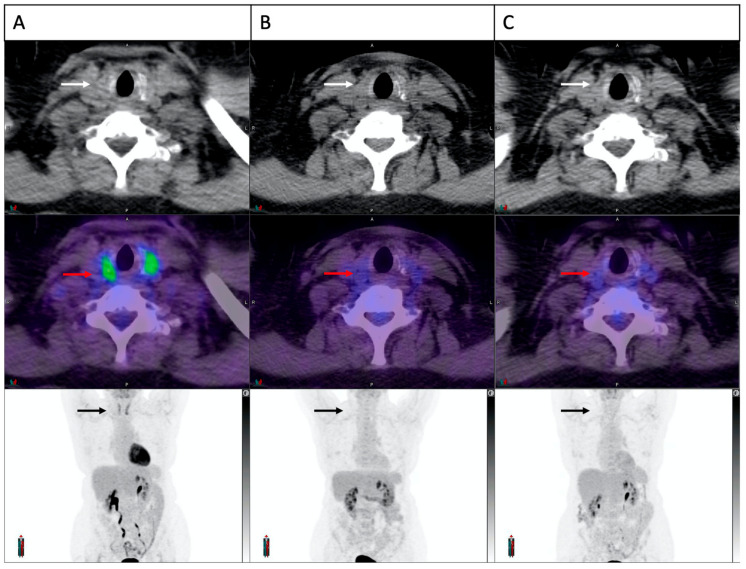
Increased thyroid FDG uptake at baseline in a patient who developed cICI related thyroiditis. Trans-axial and maximum intensity projection images of the thyroid gland demonstrated in sequence on CT scan (white arrows), fusion image (red arrows) and PET scan (black arrows). A clear increase in thyroid FDG uptake is demonstrated at baseline (**A**), which had resolved completely on both on-treatment scans at 6 and 14 weeks (**B**,**C**). The patient was euthyroid at baseline and developed biochemical thyroiditis 10 days after commencement of cICI.

**Figure 7 cancers-15-05803-f007:**
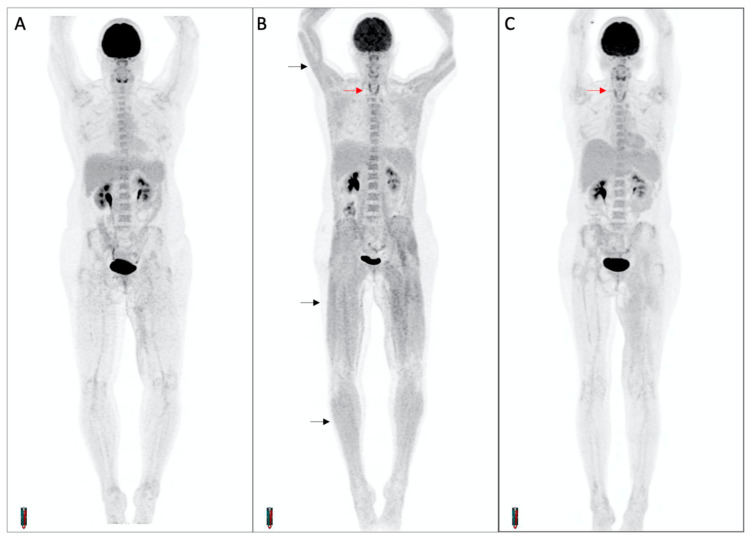
Diffuse skeletal muscle uptake of FDG in a patient with thyroiditis. Maximum intensity projection PET images at baseline (**A**) and on-treatment (**B**,**C**) demonstrate interval development of thyroiditis (red arrows) and diffuse skeletal muscle uptake (black arrows). The patient complained of progressive upper and lower limb weakness during the thyrotoxic phase, with a normal CK.

**Table 1 cancers-15-05803-t001:** Baseline clinical characteristics of patients treated with cICI.

	Thyroiditis (n = 43)	No Thyroiditis (n = 84)	*p* Value
No. (%) Male	29 (67.4%)	61 (72.6%)	0.54
No. (%) Female	14 (32.6%)	23 (27.4%)	
Age, median (IQR), years	58 (50–67)	61 (47–69)	0.99
BMI, median (IQR), kg/m^2^	28 (25–34)	27 (24–30)	0.055
No. (%) prior exposure to single-agent ICI or BRAF/MEK inhibitors			
CTLA4 inhibitor	1 (2.3%)	5 (6.0%)	0.36
PD-1 inhibitor	4 (9.3%)	30 (35.7%)	0.002
BRAF/MEK Inhibitors	25 (58.1%)	26 (31.0%)	0.003
No. (%) Prior Autoimmunity (where specified)			
≥1 autoimmune condition	3 (7.0%)	10 (11.9%)	0.39
Rheumatoid Arthritis	0 (0.0%)	2 (2.4%)	0.31
Inflammatory Bowel Disease	1 (2.3%)	1 (1.2%)	0.63
Psoriasis	0 (0.0%)	2 (2.4%)	0.31
Vitiligo	1 (2.3%)	0 (0.0%)	
Alopecia	0 (0.0%)	1 (1.2%)	
Sjogren’s Syndrome	0 (0.0%)	1 (1.2%)	
Graves’ Disease	0 (0.0%)	0 (0.0%)	
Hashimoto’s Thyroiditis	0 (0.0%)	2 (2.4%)	
Coeliac Disease	0 (0.0%)	1 (1.2%)	
Type 1 Diabetes	0 (0.0%)	1 (1.2%)	
Multiple Sclerosis	0 (0.0%)	0 (0.0%)	
Behcet’s Disease	0 (0.0%)	1 (1.2%)	
Immune Glomerulonephritis	1 (2.3%)	0 (0.0%)	
Aortitis	0 (0.0%)	1 (1.2%)	0.47
Immune-related adverse events			
No. (%) with ≥1 irAE (other than thyroiditis)	40 (90.1%)	74 (88.1%)	0.26
No. (%) with at least one grade 3/4 irAE	23 (52.3%)	35 (41.7%)	0.27
No. (%) by organ system			
Hypophysitis	9 (20.9%)	18 (21.4%)	1.0
Dermatological	22 (51.2%)	39 (46.4%)	0.71
Hepatitis	14 (32.6%)	26 (31.0%)	0.84
Enteritis/Colitis	14 (32.6%)	28 (33.3%)	1.0
Rheumatic	7 (16.3%)	8 (9.5%)	0.38
Pneumonitis	5 (11.6%)	9 (10.7%)	1.0
Nephritis	1 (2.3%)	3 (3.6%)	1.0
Myocarditis	0 (0.0%)	1 (1.2%)	1.0
CNS including eye and ear	2 (4.7%)	3 (3.6%)	1.0
Other (Lymphadenitis, Haemolytic Anaemia, Panniculitis-like T-cell Lymphoma)	1 (2.3%)	3 (3.6%)	1.0

Abbreviations: cICI, combination immune checkpoint inhibition; ICI, immune checkpoint inhibition; BMI, body mass index; CTLA4, cytotoxic T-lymphocyte-associated protein 4; PD-1, programmed death 1; BRAF, proto-oncogene B-Raf; MEK, mitogen-activated extracellular signal-regulated kinase; irAE, immune-related adverse event; CNS; central nervous system.

**Table 2 cancers-15-05803-t002:** Cross tabulation of the index test results by the results of the reference standard.

	Reference Test +	Reference Test −	Total
Index Test +	26	2	28
Index Test −	17	82	99
Total	43	84	127

**Table 3 cancers-15-05803-t003:** Cohen’s Kappa score.

	User 1: Index Test +	User 2: Index Test −	Total
User 2: Index test +	28	5	33
User 2: Index test −	3	91	94
Total	31	96	127

Agreement: 93.7% [95% CI 89.4–98.0], Cohen’s k: 0.83 (almost perfect agreement).

**Table 4 cancers-15-05803-t004:** Timing of imaging with respect to the biochemical diagnosis in patients who had confirmed thyroiditis.

	Time to Biochemical Diagnosis (Weeks)	Time to FDG-PET/CT (Weeks)	Time Interval (Weeks)
FDG-PET-detected thyroiditis N = 26	Median 2.8 (range 1.1–97)	Median 11 (range 2.6–32.1)	Median 6.35 (range 0–86.7)
No FDG-PET-detected thyroiditis N = 17	Median 3.1 (range 1.4–14.1)	Median 10.6 (range 2.7–13.1)	Median 6 (range 0.9–9.7)
*p* value	0.34	0.57	0.62

## Data Availability

Raw data may be available on request from the corresponding author. The data are not publicly available due to hospital data governance restrictions.

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
