# Peer review of "Increased Thyroidal Activity on Routine FDG-PET/CT after Combination Immune Checkpoint Inhibition: Temporal Associations with Clinical and Biochemical Thyroiditis"

_cancers, 2023, doi:10.3390/cancers15245803_

Round 1

Reviewer 1 Report

Comments and Suggestions for Authors

This is a retrospective study assessing the association of FDG-PET/CT and thyroid dysfunction in patients with melanoma being treated with combination immune checkpoint blockade.

The major limitation of this study is the timing- because the thyroiditis occurs early on treatment—it appears that the median is actually at the first cycle! The TFT changes precede the PET scan. Therefore, the imaging changes are less clinically meaningful as they do not helps with the diagnostics. Therefore, the emphasisis on showing the ROC and the sensitivity/specificity/NPV/PPV should be minimized, or at least termed by the thyroiditis being predictive of the PET changes as that is what the clinician will know first. Perhaps a focus on concordance would be a better fit. Further details on what the PET can tell us- for example that there is a higher chance of progression to overt hypothyroidism—would be appreciated.

With these caveats, I think that the overall study has the potential to add to our literature and understanding of thyroid irAEs.

Major:

·      If ultrasounds are available for these patients, were those that had PET uptake more or less likely to have nodules/ pre-existing euthyroid thyroid autoimmunity?

·      Were  patients with pre-existing hypothyroidism or Graves prior to ICI initiation included or excluded?

·      Lines 167-168—in your table it looks like prior PD-1 exposure is associated with not developing thyroiditis (perhaps these people already developed it with their first line of therapy?)

·      Lines 184-186—I am interested in the length of follow up for the patients with thyrotoxicosis that did not develop hypothyroidism subsequently and where the two subclinical patients fell. In my experience it is much more than half that will progress to hypothyroidism rather than return to euthyroidism.

·      Figure 5: I worry a bit about how to interpret the phases here is there is a long time between lab tests. I think this needs to be communicated in some way.

·      Line 264-265: overt hypothyroidism is going to be more likely to occur if patients are left longer in between doses. I am assuming that the timing of overt hypothyroidism may line up with the timing of LT4 start but I would consider looking at this as well as dose of LT4 required rather than using overt hypothyroidism as defined by TFTs.

·      Line 342-344: people that start off with thyroid Abs will almost always progress to thyroid dysfunction, so they do have some place in risk prediction. Less place in diagnostics maybe? I think it would be really interesting for you to look at the larger subset of patients that you have here that were hyperthyroid that did not become hypothyroid- did antibodies predict that? Or the speed at which someone converted from hyper to hypothyroidism?

·      Line 380-395: There is some literature of increased risk of thyroid dysfunction in other TKI- I think this is the article but there may be others as well: doi: 10.1007/s00262-022-03151-2

Minor:

·      Lines 205-207: Recognizing the temporal discordance in biochemical testing and scanning in many patients, the sensitivity, specificity, positive predictive value (PPV) and negative predictive value (NPV) were 61% [95% confidence interval (CI) 44-75], 98% [95% CI 92-100%], 93% [95% CI 77-91] and 83% [95% CI  74-90] respectively. => I am confused by what you are getting at in this sentence- as you mention the timing of the TFTs and PET don’t correlate. Are these numbers for the accuracy of PET abnormality to predict thyroid disease or vice versa?

·      Figure 5:

o   legend: change first PET to first ON TREATMENT PET

o   would add the number of weeks at which the thyroiditis is diagnosed to the arrow

o   Would make a line that shows the thyrotoxic phase and the hypothyroid phase instead of

·      Consider using a flowchart to describe patients

Author Response

Comment 1

This is a retrospective study assessing the association of FDG-PET/CT and thyroid dysfunction in patients with melanoma being treated with combination immune checkpoint blockade.

The major limitation of this study is the timing-The TFT changes precede the PET scan. Therefore, the imaging changes are less clinically meaningful as they do not help with the diagnostics. Therefore, the emphasise on showing the ROC and the sensitivity/specificity/NPV/PPV should be minimized, or at least termed by the thyroiditis being predictive of the PET changes as that is what the clinician will know first. Perhaps a focus on concordance would be a better fit. Further details on what the PET can tell us- for example that there is a higher chance of progression to overt hypothyroidism—would be appreciated. With these caveats, I think that the overall study has the potential to add to our literature and understanding of thyroid irAEs.

Response 1

We thank the reviewer for their time and feedback. We agree with the reviewer in their view that there is limited utility for FDG PET/CT as a first line diagnostic test for thyroiditis. In paragraph one of our discussion, we draw attention to the discordance in timing of the diagnosis (based on thyroid function tests) and the available imaging. The increased likelihood of progression to overt hypothyroidism is a more important interpretation of results and this was emphasised in our discussion and was one of our major conclusions. Another important clinical implication is that a uniform increase in PET glucose tracer uptake is likely due to thyroiditis rather than thyroidal tumour metastases.

We have added the following sentence to the discussion on paragraph on page 12 line 320: As changes in thyroidal uptake occur after the diagnosis is made by biochemical thyroid function tests the incidental finding of diffusely increased FDG uptake on routine on-treatment FDG PET/CT is highly diagnostic of thyroiditis rather than thyroidal tumour metastases.

We have modified our conclusions on page 13 line 415 as follows: The primary role of FDG-PET/CT is monitoring of tumoral response and not for the evaluation of thyroiditis. While we admit the limitations in utility of FDG PET/CT in the initial diagnosis of thyroiditis, the study has followed the recommendations for novel test performance characteristics in the analysis of the sensitivity/specificity/NPV/PPV of FDG PET/CT with respect to the reference standard test. However, with With an increasing incidence of thyroid abnormalities detected on routine FDG-PET/CT imaging it is of great importance to understand the implications of these findings when interpreting the FDG PET/CT. To our knowledge, this is the first study to systematically evaluate the temporal profile of FDG-PET/CT findings of thyroiditis in relation to the biochemical and clinical course of thyroid dysfunction after cICI.

We have also amended our conclusion on Page 14 line 430 as follows:  Although routine FDG-PET/CT is commonly performed after the biochemical thyroid abnormality was detected, our study demonstrates that FDG-PET/CT has a high specificity and positive predictive value for ICI related thyroiditis and is predictive of correlates with progression to permanent hypothyroidism.

Comment 2

If ultrasounds are available for these patients, were those that had PET uptake more or less likely to have nodules/ pre-existing euthyroid thyroid autoimmunity?

Response 2

Surveillance thyroid ultrasound in euthyroid patients before or after treatment with immune checkpoint inhibitors is not recommended in local or international guidelines so unfortunately this additional information was not routinely available for our study. Nodules in theory may be detected on baseline CT component, or with focal glucose tracer uptake if they are hyperfunctioning. The pattern of FDG uptake seen in our cohort was uniform and not nodular, with temporal evolution after treatment in keeping with thyroiditis. With respect to pre-existing euthyroid auto-immunity, thyroid auto-antibodies were not routinely assessed before treatment. We agree that this would be very useful information and an important future direction for biomarker investigations. We have discussed post-treatment antibody results in discussion paragraph 4.1, page 13 line 348. We have added the following review of baseline thyroid antibodies in the literature to our discussion on page 13 line 352: In our study, baseline thyroid antibody levels were not measured. Muir et al have demonstrated across several studies that baseline TPOAb and TGAb positivity were more common in patients who developed thyroid irAEs than in those who did not, and patients where there was a greater than 50% increase in baseline antibody titre or new seroconversion were similarly more likely to experience thyroiditis than those who did not[22, 25]. In thyroid irAE, antibody seroconversion may be a late or bystander response to a rapid pathogenic T cell autoimmunity, or antibody specificity in thyroid irAE may be different to that of spontaneous thyroiditis. The heterogeneity of the antibody response after treatment likely negates them as a clinically useful biomarker, however baseline antibody levels may be useful for risk prediction.

Comment 3

Were patients with pre-existing hypothyroidism or Graves prior to ICI initiation included or excluded?

Response 3

Prior autoimmunity is outlined in table 1 on page 5 line 176. No patients in our cohort had a history of Graves’ disease. 2 patients with a documented history of Hashimoto’s disease were included in the study. It is known that patients with Hashimoto’s disease may still be susceptible to acute thyroiditis, for example in the post-partum phase, and as such these two patients were not excluded. Neither developed biochemical or PET thyroiditis.    

Comment 4

Lines 167-168—in your table it looks like prior PD-1 exposure is associated with not developing thyroiditis (perhaps these people already developed it with their first line of therapy?).

Response 4

There were 7 patients excluded from analysis due to a history of checkpoint inhibitor related thyroiditis from a previous line of therapy (page 2 line 94). In the remaining cohort, prior treatment with PD-1 or CTLA-4 antibody was not associated with increased risk of developing thyroiditis following combination CPI. It is possible the patients may have had transient undetected thyroiditis with prior therapy but there was no increased risk of recurrent thyroiditis. The following comment has been added to the discussion on page 14 line 412: Of note, prior treatment with CTLA-4 inhibitiors or PD-1 inhibitors did not appear to prime the thyroid for thyroiditis events after cICI.

Comment 5

Lines 184-186—I am interested in the length of follow up for the patients with thyrotoxicosis that did not develop hypothyroidism subsequently and where the two subclinical patients fell. In my experience it is much more than half that will progress to hypothyroidism rather than return to euthyroidism.

Response 5

We thank the reviewer for pointing out that follow-up was not outlined in the text. The following line has been added to the methods section on page 3 line 98: Clinical events were followed for a minimum of 12 months. The swimmer plot demonstrates that the hypothyroid phase usually occurred within 3 months.  Frequency of biochemical monitoring of asymptomatic patients generally is reduced after the first 6 months of therapy, and it is possible that some episodes of late hypothyroidism may have been detected outside of our centre in the primary care setting.

We have added the following text to our discussion on page 13 line 329: Our finding that 53.5% of patients developed permanent hypothyroidism is similar to the published literature. In one study of 1246 patients treated with checkpoint inhibitors, permanent hypothyroidism requiring levothyroxine replacement occurred in 43% of patients who initially presented with overt thyrotoxicosis and only 3% of those who’s initial presentation was subclinical thyrotoxicosis.

Comment 6

Figure 5: I worry a bit about how to interpret the phases here is there is a long time between lab tests. I think this needs to be communicated in some way.

Response 6

In our methods section on Page 3 line 101 we outline that TFTs are performed at baseline, at every cycle for the first 4 cycles, every 4-6 weeks thereafter, or earlier in the case of clinical suspicion. The swimmer plot only stamps 3 time points: start of thyrotoxic phase, end of thyrotoxic phase and start of hypothyroid phase. The following sentence has been added to the figure footnotes on page 10 line 261 to provide clarity to the figure: Thyroid function tests were performed at baseline, every cycle for the first 4 cycles and every 4-6 weeks thereafter, or more frequently if clinically indicated.

Comment 7

Line 264-265: overt hypothyroidism is going to be more likely to occur if patients are left longer in between doses. I am assuming that the timing of overt hypothyroidism may line up with the timing of LT4 start but I would consider looking at this as well as dose of LT4 required rather than using overt hypothyroidism as defined by TFTs.

Response 7

All patients had regular monitoring of TFTs every cycle for 4 cycles then every 4-6 weeks or earlier if clinically indicated. All patients were commenced of LT4 treatment at the time of identification of biochemical hypothyroidism and as discussed on page 6 line 188, withdrawal of LT4 was not attempted. Patients are generally commenced on 50-75mcg initially, rather than full replacement (closer to 1.6mcg per kg body weight). Final thyroxine dose after a period of titration was not collected in the chart review but I agree that this would be a useful way of defining overt hypothyroidism, as would a trial of thyroxine withdrawal.    

Comment 8

Line 342-344: people that start off with thyroid Abs will almost always progress to thyroid dysfunction, so they do have some place in risk prediction. Less place in diagnostics maybe? I think it would be really interesting for you to look at the larger subset of patients that you have here that were hyperthyroid that did not become hypothyroid- did antibodies predict that? Or the speed at which someone converted from hyper to hypothyroidism?

Response 8

We agree with the reviewer’s comment. Measurement of baseline thyroid antibodies has been done by other groups, but was not available in our study. To address this comment, we have made the following changes on Page 13 line 352:

One observational study has shown that TgAb but not TPOAb positivity is associated with an increased likelihood of progression to permanent hypothyroidism following the hyperthyroid phase[24]. In our study, baseline thyroid antibody levels were not measured. Muir et al have demonstrated across several studies that baseline TPOAb and TGAb positivity were more common in patients who developed thyroid irAEs than in those who did not, and patients where there was a greater than 50% increase in baseline antibody titre or new seroconversion were similarly more likely to experience thyroiditis than those who did not[22, 25]. In thyroid irAE, antibody seroconversion may be a late or bystander response to a rapid pathogenic T cell autoimmunity, or antibody specificity in thyroid irAE may be different to that of spontaneous thyroiditis. The heterogeneity of the antibody response after treatment likely negates them as a clinically useful biomarker, however baseline antibody levels may be useful for risk prediction.

Comment 9

Line 380-395: There is some literature of increased risk of thyroid dysfunction in other TKI- I think this is the article but there may be others as well: doi: 10.1007/s00262-022-03151-2

Response 9

This is an interesting point made by the reviewer. We are aware that TKI can cause thyroiditis but in our cohort there were no patients treated with concurrent TKI. As such, to maintain simplicity we have not amended the text.  

Comment 10

Lines 205-207: Recognizing the temporal discordance in biochemical testing and scanning in many patients, the sensitivity, specificity, positive predictive value (PPV) and negative predictive value (NPV) were 61% [95% confidence interval (CI) 44-75], 98% [95% CI 92-100%], 93% [95% CI 77-91] and 83% [95% CI  74-90] respectively. => I am confused by what you are getting at in this sentence- as you mention the timing of the TFTs and PET don’t correlate. Are these numbers for the accuracy of PET abnormality to predict thyroid disease or vice versa?

Response 10

Given the frequency of thyroidal uptake changes seen on FDG PET/CT, we felt it was worthwhile to compare the accuracy of this test to the reference standard TFTs. It was important to discuss the discordance between timing of the biochemical and imaging tests, as this is a likely explanation for why not all patients with biochemical thyroiditis had an increase in thyroidal uptake on PET. The emphasis is to raise awareness that incidental findings of diffuse FDG uptake in the thyroid on routine response monitoring FDG PET/CT after combination immune checkpoint inhibition should not be dismissed and are highly specific for thyroiditis.

Comment 11

Figure 5:

o   legend: change first PET to first ON TREATMENT PET-

o   would add the number of weeks at which the thyroiditis is diagnosed to the arrow

o   Would make a line that shows the thyrotoxic phase and the hypothyroid phase instead of

  • Consider using a flowchart to describe patients

We thank the reviewer for this feedback. With regard to the first two points, the figure legend has been adjusted as follows:

Figure 5. Swimmer Plot demonstrating the timing of the first on-treatment FDG-PET/CT with respect to the thyroiditis diagnosis, typically represented by an early thyrotoxic phase followed by euthyroid and/or hypothyroid phase. Thyroid function tests were performed every 4-6 weeks or more frequently if clinically indicated. The horizontal black arrow represents an outlier, whereby the thyrotoxicosis was diagnosed 97 months after the first FDG-PET/CT.

Regarding the second two points, we discussed multiple pictorial ways to tell the story simply and found that lines and flow charts became very difficult to follow. We feel the swimmer plot to be the best way we can demonstrate the phases of biochemical thyroid abnormalities and the timing of the PET scan in one diagram, and we hope this figure can remain in our manuscript.

Reviewer 2 Report

Comments and Suggestions for Authors

The early detection and the identification of thyroiditis by the utilization of the FDG-PET/CT imaging system is a novel and interesting technique. However, the timing of the imaging   FDG-PET/CT in comparison with the stage of the disease is a critical issue that should be taken under consideration.  

The present manuscript is a well written text that describe a novel approach that could have a clinical use. The utilization of a larger size of patients’ dataset in different time points it could be a valuable clinical contribution.  

Author Response

Comment 1

The early detection and the identification of thyroiditis by the utilization of the FDG-PET/CT imaging system is a novel and interesting technique. However, the timing of the imaging   FDG-PET/CT in comparison with the stage of the disease is a critical issue that should be taken under consideration.  

Response 1

It is true that the first on-treatment PET was performed significantly later than the biochemical test which made the diagnosis of thyroiditis. We have admitted this is a limitation and a potential reason that some patients with biochemical thyroiditis did not have PET changes.

We have modified our conclusions on page 13 line 415 as follows: The primary role of FDG-PET/CT is monitoring of tumoral response and not for the evaluation of thyroiditis. While we admit the limitations in utility of FDG PET/CT in the initial diagnosis of thyroiditis, the study has followed the recommendations for novel test performance characteristics in the analysis of the sensitivity/specificity/NPV/PPV of FDG PET/CT with respect to the reference standard test. However, with With an increasing incidence of thyroid abnormalities detected on routine FDG-PET/CT imaging it is of great importance to understand the implications of these findings when interpreting the FDG PET/CT. To our knowledge, this is the first study to systematically evaluate the temporal profile of FDG-PET/CT findings of thyroiditis in relation to the biochemical and clinical course of thyroid dysfunction after cICI.

Comment 2

The present manuscript is a well written text that describe a novel approach that could have a clinical use. The utilization of a larger size of patients’ dataset in different time points it could be a valuable clinical contribution. 

Response 2

We appreciate this comment and agree that further analysis of larger sample size is valuable.  

Reviewer 3 Report

Comments and Suggestions for Authors

Authors show an interesting and original  report of FDG PET  patterns of uptake in thyroid tissue during ciCI.The manuscript is interesting and clearly provides a pictorial review associated with a good description of the pathophysiology of thyroiditis as well as charactristics and patterns of uptake in thyroid tissue in patients with thyroidits developped during immunotherapy

The increased incidence of thyroiditis in patients  with previous treatments with BRAF/MEK inhibition is also interesting and well commented. 25 patients with thyroidits and FDG PET had previous BRAF/MEK treatment. How many of them had positive FDG PET and mainly in which phase of the thyroiditis? In all of these patients thyroidits was confirmed or discovered after initiation of cICI?

There were only 4 patients with thyroiditis  out of 34 patients with previous PD-1 inhibitors. Could authors also comment briefly on this aspect? 

Patients describe  Se, Sp, PPV and NPV for patients  with positive FDG PET and confirmed thyroiditis.

However , I think it is important that authors provide analysis of Se Sp, PPV and NPV  of the FDG PET to correctly identify thyroiditis as FDG PET was positive in 26/43 patients with thyroiditis

 As FDG PET is realised all 12 wks  as long as thyroiditis occurs in the 3 wks following ciCI onset, it is not surptising that FDG uptake is noticed mainly in th hypothyroid phase of the thyroiditis.

 In Figure 5 it seems that there was a tendency for  more patients  with thyrotoxic phase of their thyroiditis  to have a negative PET/CT. Were something particular as previous other immune  AE s in these patients compared to patients with thyroidits and positive FDG PET?

Could authors try to explain and comment this aspect?

I think it is difficult to say that positive FDG-PET predicts progression to permanent hypothyroidism as long as the FDG PET is realised in a late phase of the natural history of thyroditis.

 The manuscript is interesting and well written and provides a good description of the patterns of thyroiditis induced bu cICI.Although this aspect dose not provide clinical impacts as TFTs are regularly performed in patients on cICI.

Comments on the Quality of English Language

Excellent

Author Response

Comment 1

Authors show an interesting and original report of FDG PET patterns of uptake in thyroid tissue during ciCI.The manuscript is interesting and clearly provides a pictorial review associated with a good description of the pathophysiology of thyroiditis as well as charactristics and patterns of uptake in thyroid tissue in patients with thyroidits developped during immunotherapy

The increased incidence of thyroiditis in patients with previous treatments with BRAF/MEK inhibition is also interesting and well commented. 25 patients with thyroidits and FDG PET had previous BRAF/MEK treatment. How many of them had positive FDG PET and mainly in which phase of the thyroiditis? In all of these patients thyroiditis was confirmed or discovered after initiation of cICI?

Response 1

We thank the reviewer for this comment. The focus of our paper was to look at thyroid irAEs in patients receiving combination CTLA4/PD1 inhibition. We demonstrate in figure 5, that in many cases the PET was performed after the end of the thyrotoxic phase. As the BRAF/MEK inhibition was a prior line of therapy (not in combination) we are concerned that analysing the presence and timing of FDG PET changes in patients previously treated with BRAF/MEK inhibition (but not receiving current treatment) may not reflect a causal relationship or association. Regarding the last question, thyroiditis was discovered after initiation of cICI. Patients diagnosed with thyroid irAE from a prior line of treatment were excluded (page 2 line 94).

Comment 2

There were only 4 patients with thyroiditis  out of 34 patients with previous PD-1 inhibitors. Could authors also comment briefly on this aspect? 

Response 2

The following comment has been added to the discussion on page 14 line 412: Of note, prior treatment with CTLA4 inhibitiors or PD-1 inhibitors did not appear to prime the thyroid for thyroiditis events after cICI.

Comment 3

Patients describe  Se, Sp, PPV and NPV for patients  with positive FDG PET and confirmed thyroiditis.

However , I think it is important that authors provide analysis of Se Sp, PPV and NPV  of the FDG PET to correctly identify thyroiditis as FDG PET was positive in 26/43 patients with thyroiditis. As FDG PET is realised all 12 wks  as long as thyroiditis occurs in the 3 wks following ciCI onset, it is not surptising that FDG uptake is noticed mainly in th hypothyroid phase of the thyroiditis.

Response 3

The reviewer is correct in that not all patients with thyroiditis had abnormal PET findings, which we have discussed. We state the performance characteristics of FDF PET/CT in the diagnosis of thyroiditis with respect to the reference standard biochemical TFTs, while pointing out the major limitation being the discordance in timing of the first on-treatment FDG PET/CT. FDG PET/CT was performed per institutional protocol for response monitoring every 12 weeks. We want to make sure that diffuse FDG PET uptake in the thyroid is not dismissed by interpreting physicians and be reported as such.  Could we please request further clarification of this question if our response does not address the reviewer’s concerns.

Comment 4 

In Figure 5 it seems that there was a tendency for more patients with thyrotoxic phase of their thyroiditis to have a negative PET/CT. Were something particular as previous other immune  AE s in these patients compared to patients with thyroidits and positive FDG PET?

Could authors try to explain and comment this aspect?

Response 4

The reviewer points out that amongst patients with a negative PET, 12/17 (70.6%) had a thyrotoxic phase compared to 14/26 (54%) of those with a positive PET. Another way of looking at the data which is more easily understood, is to look at proportion of PET abnormalities amongst patients with and without a thyrotoxic phase. In this case, 54% with a thyrotoxic phase had a positive PET and 46% with a thyrotoxic phase had a negative PET. As we have discussed on page 13 line 329, an initial thyrotoxic phase is a risk factor for permanent hypothyroidism, and as we found FDG PET uptake to also predict hypothyroidism, these figures seem appropriate.

A prior history of autoimmunity with respect to biochemical thyroiditis is outlined in table 1 (page 5 line 176). As the absolute number of patients with prior autoimmunity was small in the thyroiditis group (n=5), we did not expand this analysis to the PET positive group, which was smaller again than the biochemical thyroiditis group.

Comment 5

I think it is difficult to say that positive FDG-PET predicts progression to permanent hypothyroidism as long as the FDG PET is realised in a late phase of the natural history of thyroditis.

Response 5

We thank the reviewer for pointing out that in many cases, the PET scan was performed after the hypothyroidism occurred. As such, we have amended our conclusion on Page 14 line 429 as follows:  

Although routine FDG-PET/CT is commonly performed after the biochemical thyroid abnormality was detected, our study demonstrates that FDG-PET/CT has a high specificity and positive predictive value for ICI related thyroiditis and is predictive of correlates with progression to permanent hypothyroidism.

Comment 6

The manuscript is interesting and well written and provides a good description of the patterns of thyroiditis induced by cICI.Although this aspect dose not provide clinical impacts as TFTs are regularly performed in patients on cICI.

We appreciate this feedback. To emphasise the role of our findings and the limitations, the conclusion has been modified on Page 14 line 415 as follows:

While we admit the limitations in the utility of FDG PET/CT in the initial diagnosis of thyroiditis, the study has followed the recommendations for novel test performance characteristics in the analysis of the sensitivity/specificity/NPV/PPV of FDG PET/CT with respect to the reference standard test. With an increasing incidence of thyroid abnormalities detected on routine FDG-PET/CT imaging it is of great importance to understand the implications of these findings when interpreting the FDG PET/CT.